# The Incidence of Contrast-Induced Nephropathy Among Low-Risk Cancer Patients with Preserved Renal Function on Active Treatment Undergoing Contrast-Enhanced Computed Tomography: A Single-Site Experience

**DOI:** 10.3390/healthcare14010115

**Published:** 2026-01-03

**Authors:** Ahmad Subahi, Nada Alhazmi, Maryam Lardi, Fatimah Alkathiri, Layan Bokhari, Sultanah Alqahtani, Nesreen Abourokba, Khalid Alshamrani

**Affiliations:** 1Basic Sciences, College of Science and Health Professions, King Saud Bin Abdulaziz University for Health Sciences, Jeddah 21428, Saudi Arabia; 2King Abdullah International Medical Research Center, Jeddah 12211, Saudi Arabia; 3Ministry of the National Guard—Health Affairs, Jeddah 22384, Saudi Arabia; 4College of Applied Medical Sciences, King Saud bin Abdulaziz University for Health Sciences, Jeddah 21423, Saudi Arabia; 5College of Medicine, King Saud bin Abdulaziz University for Health Sciences, Jeddah 22490, Saudi Arabia

**Keywords:** contrast-induced nephropathy, contrast-enhanced computed tomography, oncologic imaging, iodinated contrast media, cancer patients, serum creatinine, estimated glomerular filtrate rate, renal function, incidence, contrast safety

## Abstract

**Background/Objectives**: Contrast-induced nephropathy (CIN) is a common iatrogenic or medically induced condition among patients who receive intravenous infusion of iodinated contrast media that can cause renal insufficiency, raise the cost of care, and increase mortality risk. This study evaluated the incidence of CIN and predictors of renal function among cancer patients receiving contrast-enhanced computed tomography (CECT). **Methods**: A prospective, single-center longitudinal study was conducted at King Abdul-Aziz Medical City’s (Jeddah) medical imaging department from December 2021 to December 2023. Convenience sampling was used to select patients who were exposed to CECT based on data filled in the electronic medical record during the study period. **Results**: The final sample constituted 80 patients (47.71% attrition, mean age = 55.5 years, 58.75% male). The high attrition rate was associated with participants with incomplete records, those who were lost to follow-up, and those whose follow-up Scr was collected after 72 h from CECT administration. There was no statistically significant change in Scr following contrast exposure (mean increase 0.9 µmol/L; paired t = 1.41, *p* = 0.162; Wilcoxon *p* = 0.326). The incidence of CIN was 3.75% (3 of 80 patients; 95% confidence intervals (CI), 1.28–10.39%). Regression analysis showed no statistically significant associations between the percentage change in Scr and age, sex, baseline creatinine, or eGFR category (model R^2^ = 0.07). No clinically meaningful predictors of CIN were identified. **Conclusions**: The incidence of CIN in this study’s cohort of low-risk cancer patients undergoing CECT was low, and contrast exposure did not produce significant short-term changes in renal function. These findings support the safety of modern contrast agents in oncology imaging, but multi-center studies with larger samples and more robust methods are warranted to refine CIN risk assessment in cancer patients undergoing CECT.

## 1. Introduction

Contrast-induced nephropathy (CIN) is a common iatrogenic or medically induced condition among residents of the Kingdom of Saudi Arabia (KSA) that is associated with high costs of care, renal insufficiency, and mortality [1,2,3]. This condition results from intravenous infusion of iodinated contrast media [1]. CIN is clinically defined as at least a 0.5 mg/dL or 25% increase in serum creatinine (Scr) from pre-administration within 24 to 72 h of contrast media administration [1,2]. Other scholars define CIN as a subclinical, acute form of renal failure that results in gradual worsening of renal function within 48–72 h [1,3]. Iodinated contrast media are often administered to patients undergoing cancer imaging to highlight body structures and enhance the detection of abnormalities in what is referred to as contrast-enhanced computed tomography (CECT) [1,2]. CECT is a well-established imaging technique that is preferred when screening different types of cancer due to its high sensitivity and specificity [4]. However, the safety of this test is widely debated due to its likelihood of raising Scr levels and predisposing patients to CIN [1,2,3].

Safe administration of intravenous contrast media during CECT requires Scr levels of 1.5 mg/dL or lower since values above this threshold could lead to renal insufficiency [5]. Despite the difficulty of documenting data about CIN due to its asymptomatic nature and inconsistencies in definition, many scholars have attempted to document the condition’s incidence among persons undergoing CECT under various conditions. One study found that the incidence of CIN is 8.0% in patients with near-typical Scr levels [2]. Other studies have documented incidences that range from less than 5% [1,5] among patients with normal renal function to 12–27% [2] among those with preexisting renal impairment, and 50% [3] among those with diabetes and severe renal injury. Studies have documented various risk factors associated with CIN. These differences show a correlation between CIN and endocrinologic conditions like renal impairment and diabetes [1,2,3]. For instance, patients with a baseline Scr that exceeds 1.5 mg/dL are at a higher risk of developing CIN [5]. The risk of developing contrast-induced nephropathy (CIN) is influenced by several interrelated patient- and treatment-related factors. Baseline renal vulnerability is a primary determinant, with patients who have impaired kidney function or underlying endocrinologic disorders being at particularly high risk [2,3]. Treatment-related factors, especially the use of chemotherapeutic agents during contrast media administration, further increase susceptibility to CIN. In addition, demographic characteristics, including advanced age and female sex, have been identified as contributory risk factors in some studies [2]. Finally, a range of comorbid medical conditions, such as congestive heart failure, hypotension, dehydration, cirrhosis, atherosclerosis, and anemia, have been consistently associated with an elevated risk of CIN [2].

CIN is an asymptomatic transient decline in renal function that requires laboratory tests like Scr to facilitate detection [1,2]. However, this condition leads to severe hospital-acquired renal insufficiency in 11% of patients, the need for hemodialysis, or even death [1,4,5]. Patients who develop CIN may also experience fluid overload, pulmonary edema, and increased risk of cardiovascular events [2,3]. CIN also prolongs hospitalization, increases treatment costs by an average of USD 10,345, raises costs by up to USD 15,900 among those who develop kidney damage, and raises total all-cause mortality by up to 24% [6,7,8].

Unfortunately, there are several gaps in the current literature on the incidence of CIN among cancer patients on active treatment undergoing CECT. First, the inconsistency in clinical definitions may lead to differences in the reported incidence of CIN [1,2]. Second, the asymptomatic nature of CIN means that many of these cases are likely to be missed unless confirmed clinically [1,3]. Third, many studies also fail to account for confounders like the several risk factors associated with CIN as stated above. Fourth, there is a scarcity of regional epidemiology data, particularly within KSA. As such, this longitudinal study examines the incidence of CIN among adults diagnosed with cancer undergoing CECT in Jeddah, KSA.

## 2. Materials and Methods

### 2.1. Participants and Procedures

A prospective, single-center longitudinal study of adult patients with cancer who received CECT between December 2021 and December 2023 was performed to gather and analyze data to determine the incidence of CIN during CECT. The study targeted patients who received oncology imaging and treatment at the medical imaging department at King Abdulaziz Medical City, Ministry of National Guard Affairs, Jeddah [9]. A longitudinal design was selected since it necessitates repeated observations or temporal follow-up of the same subjects [10]. This design is thus valuable for studying the incidence of CIN among cancer patients on active treatment undergoing CECT in Jeddah between 2021 and 2023 [10]. Purposive sampling was used to intentionally select the most relevant sample based on a set of inclusion/exclusion criteria [10]. It was determined that a sample of 133 patients could be sufficient to achieve a power of 0.80 at *p* < 0.05 [10].

### 2.2. Data Collection and Management

Data for all patients who met the inclusion criteria for this study were extracted from the facility’s electronic medical record (EMR). All eligible patients’ electronic medical records were obtained for the assessment using the Health Information System “BestCare2”. All participants were exposed to iohexol (Omnipaque), which is a low-osmolar contrast agent commonly used in CECT within the study site. Iohesol (Omnipaque), is manufactured by GE health care; it was sourced from Saudi Arabia Drug Store Company, Jeddah, which is its distributor in Saudi Arabia. The inclusion criteria included adults (aged 18 years and older), all genders, persons already diagnosed with cancer, persons on active treatment, persons whose baseline estimated glomerular filtrate rate (eGFR) was not recorded, individuals with a baseline Scr of 1.5 mg/dL or below, and persons who underwent CECT. Baseline eGFR and Scr levels were selected since they align with the recommended safety thresholds for administering contrast media [5,11]. Data for all patients who did not meet these criteria, including pediatric patients, persons with renal impairment, kidney transplant recipients, those on hemodialysis, or those who did not complete their scheduled follow-up, were excluded from the study dataset. The collected data included demographic variables of age and gender, visit type categorized as either inpatient (IP) or outpatient (OP), CECT date, type of diagnosed cancer, baseline Scr, follow-up Scr, and baseline eGFR. Baseline Scr was defined as the Scr level before CECT, while follow-up Scr was defined as the Scr level collected between 24 and 72 h after CECT [1,3]. Normal eGFR was defined as levels equal to or above 60 mL/min; the level was categorized as abnormal [11]. The study’s endpoint was CIN, which was operationalized as a 25% increase in Scr levels above baseline within 24 to 72 h after CECT due to its high sensitivity to subtle changes in Scr, which may happen within the selected low-risk sample [1,3].

### 2.3. Ethical Considerations

This study required an institutional review board (IRB) approval. As such, IRB approval was obtained from King Abdullah International Medical Research Center before any data was gathered or analyzed. Additionally, approval from the radiology department head was obtained before the study was conducted. Patients were assigned reference numbers for anonymity, while confidentiality was maintained by linking each exam to the assigned reference number. All datasets were deidentified, compiled in a Microsoft Excel spreadsheet, and stored on an encrypted drive to protect data privacy [10].

### 2.4. Statistical Analysis

The Excel spreadsheet was then exported to Stata version 19 [12] for statistical analysis. Continuous variables were presented as mean ± standard deviation (SD) or median and interquartile range (IQR) if normal or skewed, respectively. On the other hand, categorical variables were reported as frequencies and percentages. Two-sided *p* < 0.05 was considered statistically significant for all measures [10]. Distributions were examined with histograms and Shapiro–Wilk tests to assess normality, while paired comparisons of pre/post-contrast Scr were performed using paired *t*-tests and Wilcoxon signed-rank tests otherwise. Incidence proportions with 95% confidence intervals (CI) were calculated for binary CIN outcomes. The study estimated associations between potential predictors of age, sex, baseline eGFR, cancer type, visit type (IP vs. IP), and CIN using logistic regression. The multivariable modeling for this study included age, sex, baseline eGFR (continuous), and cancer type (categorical). Model fit and multicollinearity were also assessed to check variance inflation factors.

## 3. Results

A total of 158 cancer patients received CECT during the study period between December 2021 and December 2023. However, only 80 records were complete (mean age = 55.5 years, 58.75% male), indicating a 47.71% attrition rate. The high attrition rate was associated with participants with incomplete records, those who were lost to follow-up, and those whose follow-up Scr was collected after 72 h from CECT administration. The sample represented 15 (18.75%), 31 (38.75%), and 34 (42.50%) persons aged 18–40 years, 41–60 years, and 61–100 years, respectively. Additionally, 80% of the patients had normal eGFR, while the values in 20% of participants were lower than 60 mL/min. More than half of these patients (53.75%) received inpatient care. Breast, colon, and lung were the most common types of cancer at 23.75%, 11.25%, and 11.25%, respectively. A majority of participants (85%) had their Scr levels assessed in 48–72 h after CECT, while the remaining (15%) participants had their levels assessed between 24 and 72 h after CECT. Table 1 below shows a summary of these descriptive statistics as well as the percentage of patients diagnosed with other malignancies.

There was significant variability in the change (increase or decrease) in Scr post-contrast vs. baseline. In this case, the mean baseline Scr pre-contrast was 79.4 µmol/L compared to 80.3 µmol/L post-contrast. As such, the mean Scr increased by approximately 0.9 µmol/L after exposure to contrast media. Figure 1 below is a histogram illustrating these changes. However, a paired t-test showed t = 1.41 (*p* = 0.162), which indicated that the 0.9 µmol/L mean increase in Scr post-contrast exposure was not statistically significant. On the same note, a Wilcoxon signed-rank test suggested no significant median shift in Scr after contrast (*p* = 0.326). These findings indicated that the administration of contrast media did not produce a meaningful increase in Scr among cancer patients following CECT.

The analysis also confirmed that only 3 cases met criteria for diagnosing CIN or a 25% increase in Scr post-contrast exposure. The figure yielded an incidence of 3.75% (95% CI: 1.28–10.39%), which suggested that clinically significant Scr or nephrotoxicity was uncommon in the sample. Figure 2 below illustrates the changes in baseline vs. post-contrast Scr and highlights the three cases with a slightly higher than 25% increase in Scr.

Predictors of renal function alteration were identified through a multivariable linear regression performed using the percentage change in Scr as a continuous outcome using age, sex, baseline Scr, and categorized eGFR (normal vs abnormal) as independent variables. The ordinary least square (OLS) regression showed that age (0.05% increase per year, *p* = 0.41)), sex (1.80% higher in males gender, *p* = 0.29), baseline Scr (0.03% decrease per µmol/L, *p* = 0.18), and eGFR (4.70% higher in abnormal eGFR, *p* = 0.09) did not predict Scr change. Table 2 below represents a summary of correlation coefficients for each of these variables and their corresponding *p*-values. While each of these variables raised or lowered the risk of Scr rise, none of the associations was statistically significant. Overall, the model explained little variation in the outcome (R^2^ = 0.07), which indicated no strong predictors of CIN from the sample.

## 4. Discussion

This study included 80 adults with cancer undergoing CECT. The sample revealed a very low incidence of CIN defined as a ≥25% increase in serum creatinine [1,3]. Notably, only three patients (3.75%) met the specified criterion for diagnosing CIN. However, the incidence may differ because the definition of CIN has changed from a relative increase in Scr to an absolute increase (0.5 mg/dL) or when the duration of observation is changed, as is the case with some studies that exclude the duration between 24 and 48 h [1,2,3]. Further paired analyses did not demonstrate any statistically significant increase in Scr after oncology patients were exposed to iodinated contrast media. On the same note, multivariable modeling failed to identify strong or significant predictors of Scr change measured as a percentage, which suggested that contrast exposure had minimal measurable impact on CIN incidence within the selected cohort. An interesting finding is that eGFR yielded the greatest influence on Scr compared to other tested predictors (see Table 2 above). Figure 3 below further validates the significant increase in Scr post-contrast vs. baseline (4.7%). This increase is consistent with the definition of CIN as a decline in kidney function [2]. However, the trend was not statistically significant since its *p*-value of 0.09 was slightly higher than the threshold of <0.05. A similar pattern was reported in a prospective cohort study of 410 patients, which found a slight but statistically insignificant increase in Scr among patients with higher eGFR [11].

The relationship between contrast exposure and the incidence of CIN reported in this study aligns with other findings reported in the current literature. For instance, a large observational study of 820 cancer patients with a baseline Scr ≤ 1.5 mg/dL reported a CIN incidence of 8.0%, either a ≥25% or a ≥0.5 mg/dL increase in Scr as CIN definition [13]. The same study identified serial computed tomography exams, hypotension, and peritoneal carcinomatosis as predictors of CIN [13]. Similarly, a larger observational study of 160 adults found that the incidence of CIN was 8.8%, although the authors did not find a correlation between age and gender [14]. The 8.0–8.8% incidence is significantly higher than the 3.75% reported in this study. Other studies have also reported higher incidences ranging from 5% to 27% [1,2,5] as well as an extreme of 50% in patients with comorbidities [2]. As such, the current study suggests that the risk of CIN among low-risk oncology populations with preserved renal function receiving treatment in Jeddah may be lower than historically feared. This suggestion is consistent with current literature, which shows that modern contrast agents with lower osmolarity are safer, particularly when used in low-risk populations like predominantly persons with normal eGFR [15].

However, some researchers have developed a risk prediction model for CIN among patients undergoing CECT with preventive measures. One of these studies used a larger cohort of 2240 adults with cancer and a definition of CIN as a ≥25% increase in Scr, found that the incidence of CIN was only 2.5% [16]. The study also found lower eGFR, diabetes mellitus, and low serum albumin as key predictors of CIN [16]. The authors found a moderate discriminative ability (c-statistic = 0.73), which indicated that CIN remains rare in many patients, particularly among patients with chronic kidney disease, and that CIN risks can be stratified [16]. As such, evidence from recent empirical studies [13,14,15,16] shows a stark contrast in the incidence of CIN among patients undergoing CECT as well as the factors that predict the occurrence of this condition. These inconsistencies suggest a need for further research on the topic to determine the actual relationship between contrast exposure and CIN, as well as the mediating effect of some of the most widely discussed factors like eGFR, gender, and age.

More broadly, systematic reviews and meta-analyses have consolidated evidence from empirical studies on contrast administration and its influence on CIN. One review consolidated evidence from 21 studies with a pooled sample of 169,455 and concluded that CECT may not significantly increase CIN or other renal function abnormality risks (OR = 0.97, 95% CI 0.85–1.11) in many settings [17]. Another meta-analysis that focused on patients with chronic kidney disease reported an odds ratio for CIN of 1.07 (95% CI 0.98–1.17) for those who were exposed to contrast versus those who did not, based on a pooled sample of 55,963 patients from six studies [18]. This study suggested minimal additional risk attributable to modern contrast media regardless of renal function at baseline.

Taken together, empirical evidence from this study and synthesized evidence from the literature suggest that CIN is uncommon in cancer patients with relatively preserved baseline renal function, particularly when modern preventive strategies are applied. First, the risk of CIN would naturally be lower among cancer patients with normal or near-normal base levels, indicated by Scr and eGFR levels. Second, modern contrast practices like lower-osmolar or iso-osmolar contrast agents and careful patient selection could help mitigate traditional CIN risks. Third, a relative increase (Scr increase ≥ 25%) rather than an absolute number (a fixed 0.5 mg/dL) may be more sensitive and appropriate in CIN measurement. This approach could be more effective in identifying clinically meaningful change in patients with lower baseline creatinine. While the regression analysis performed in the current study did not identify strong predictors of percent Scr change, these patterns may reflect low event rate and limited power. Nevertheless, mediators like diabetes or prior chemotherapy implicated in other studies were not considered in the current study, which may explain its different effect size compared to other studies.

### 4.1. Clinical Implications

These findings could inform risk stratification. Given the low incidence of CIN in the current study’s cohort, clinicians may consider less restrictive use of contrast media in selected cancer patients, especially if high diagnostic values are reported following contrast-enhanced imaging. These findings will also inform preventive measures like limiting repeat CECTs when possible and close post-contrast monitoring, particularly in higher-risk subgroups like those with underlying renal or endocrine complications. The findings from this study further support the utility of risk models to personalize decision-making around contrast use in oncology imaging.

### 4.2. Limitations

This study utilized only a small sample of 80 patients from an initial group of 153 patients who received CECT during the study period. Unfortunately, the high attrition rate (47.71%) exposes the findings to validity and generalizability issues [10]. Nevertheless, the regression analysis is underpowered to identify modest predictors with only three CIN events. The single-center design could also hinder the generalizability of findings to other settings, particularly those with different protocols, contrast agent types, and patient comorbidity profiles. Third, the short follow-up window fails to capture potential longer-term renal consequences. The study may also suffer residual confounding since it only adjusted for key covariates and ignored other unmeasured factors like chemotherapy timing, volume of contrast, and hydration status that may influence CIN risk by increasing the chances of post-contrast acute kidney injury, raising osmotic load, and reducing renal flow and eGFR, respectively. This study is also biased against oncology patients with baseline Scr ≤ 1.5 mg/dL based on the sample selection criteria. As such, these findings may not be applicable to this high-risk group of oncology patients. Larger, more inclusive, multi-center studies with more robust research methodologies and longer follow-up durations are thus needed to obtain more reliable findings.

## 5. Conclusions

This study found that CIN, as measured by a ≥25% Scr increase, is a relatively rare phenomenon with an incidence of only 3.75% in a cancer patient cohort of 80 adults from Jeddah with predominantly preserved renal function. The study is the first to investigate this phenomenon in Jeddah, KSA, and among the few that have extended the investigation to cover common confounders like age, gender, baseline renal function, type of clinic visit, and type of diagnosis. These findings thus call for a nuanced and balanced approach to contrast use in oncology imaging that weighs diagnostic benefits against potential renal risks. The data from this study support the safety of CECT in many cancer patients when used judiciously. 

## Figures and Tables

**Figure 1 healthcare-14-00115-f001:**
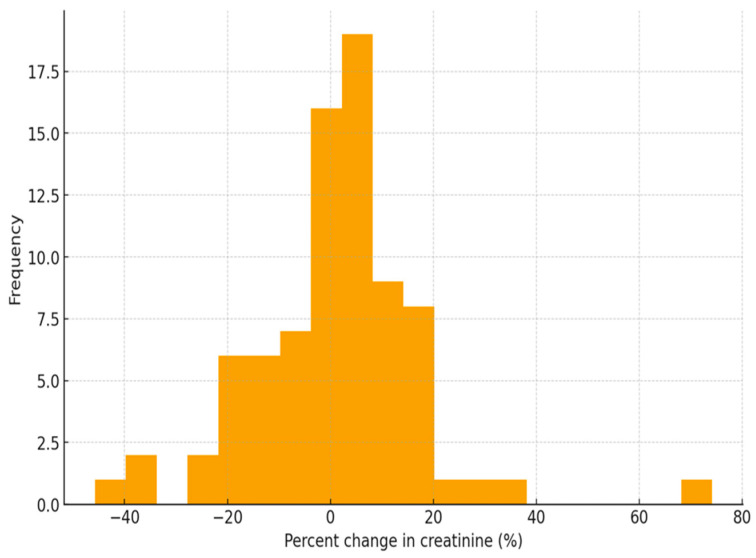
A histogram showing post-contrast vs. baseline percentage change in Scr.

**Figure 2 healthcare-14-00115-f002:**
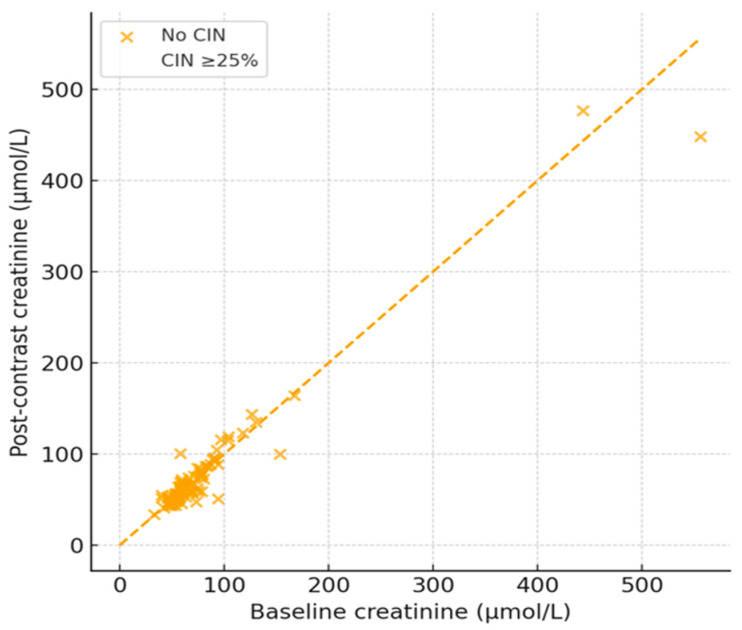
A scatter diagram showing post-contrast vs. baseline Scr.

**Figure 3 healthcare-14-00115-f003:**
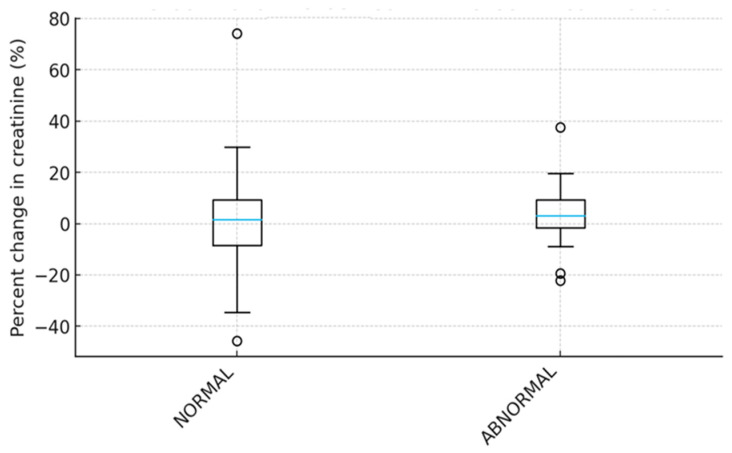
Percentage change in Scr based on eGFR category.

**Table 1 healthcare-14-00115-t001:** Summary of descriptive statistics.

Demographic Variables	Total	%
**Age**		
18–40	15	18.75%
41–60	31	38.75%
61–100	34	42.50%
**Gender**		
Female	33	41.25%
Male	47	58.75%
**eGFR level**		
Normal (≥60 mL/min)	64	80.00%
Abnormal (<60 mL/min)	16	20.00%
Type of visit		
Outpatient (OP)	37	46.25%
Inpatient (IP)	43	53.75%
**Diagnosis**		
Breast cancer	19	23.75%
Colon cancer	9	11.25%
Lung cancer	9	11.25%
Esophageal cancer	4	5.00%
Pancreatic cancer	4	5.00%
Ewing sarcoma	3	3.75%
Lymphoma	3	3.75%
Rectal cancer	3	3.75%
Renal cell carcinoma	3	3.75%
Neoplasm of the liver	2	2.50%
Neoplasm of the rectum	2	2.50%
Prostate cancer	2	2.50%
Sarcoma	2	2.50%
Anal cancer	1	1.25%
Bladder cancer	1	1.25%
Brain tumor	1	1.25%
Duodenal cancer	1	1.25%
Gallbladder adenoacanthoma	1	1.25%
Gastric cancer	1	1.25%
Hodgkin lymphoma	1	1.25%
Nasopharynx neoplasm	1	1.25%
Neoplasm of the endometrium	1	1.25%
Neuroendocrine carcinoma	1	1.25%
Osteosarcoma	1	1.25%
Skin cancer	1	1.25%
Small bowel cancer	1	1.25%
Stomach cancer	1	1.25%
Testis cancer	1	1.25%

**Table 2 healthcare-14-00115-t002:** Summary of OLS regression for model predictors.

Predictor	Coefficient	*p*-Value
Age	+0.05% per year	0.41
Male sex	1.80%	0.29
Baseline creatinine	−0.03% per µmol/L	0.18
Abnormal GFR (vs. normal)	4.70%	0.09

## Data Availability

The raw data supporting the conclusions of this article will be made available by the authors on request. The data are not publicly available due to privacy concerns.

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
