# Peer review of "The Incidence of Contrast-Induced Nephropathy Among Low-Risk Cancer Patients with Preserved Renal Function on Active Treatment Undergoing Contrast-Enhanced Computed Tomography: A Single-Site Experience"

_healthcare, 2026, doi:10.3390/healthcare14010115_

Round 1
Reviewer 1 Report
Comments and Suggestions for Authors
The manuscript could not be considered for publication in the current form for following reasons.
- Authors have presented the rationale and objectives well, but the gap for cancer patients undergoing CECT in KSA should be stated more clearly and earlier, rather in the end of the paragraph.
- Authors reported incidence ranges (less than 5%… to 12–27%… and 50% in high-risk groups) are useful but need clearer structure and context.
- Risk factors are quietly repetitive. It would be stronger if grouped conceptually (e.g., baseline kidney function, comorbidities) rather than as a long linear list.
- Some risk factors (e.g., female gender) remain controversial, it might be more appropriate to phrase these as reported in some studies.
- Calling the design retrospective cross-sectional may be confusing, as CIN is inherently a post-exposure outcome. Authors need to clarify it so that the temporal sequence is evident.
- The choice of 24–72 hours for Scr reassessment is standard but authors need to consider clarifying how many patients had creatinine checked at each time point and whether there was variation in timing that might affect detection of CIN.
- The justification for a cross‑sectional design is not accurate for CIN, which requires temporal follow-up after contrast, it should be revised, and the rationale and time window better aligned with the outcome definition.
- When stating that CIN risk may be lower than historically feared, it would be helpful to clarify this as applying to similar low‑risk oncology populations, with preserved renal function, rather than to all cancer patients.
Author Response
Comment 1- Authors have presented the rationale and objectives well, but the gap for cancer patients undergoing CECT in KSA should be stated more clearly and earlier, rather in the end of the paragraph.
Response- Thank you for pointing this out. We have made the required changes. The introduction has been edited and the problem statement is mentioned right from the first paragraph of the introduction.
Comment 2- Authors reported incidence ranges (less than 5%… to 12–27%… and 50% in high-risk groups) are useful but need clearer structure and context.
Response- Thank you for pointing this out. We have made the required changes. This has now been clarifies in the second paragraph of the introduction. Context regarding the variations in CIN incidence across studies was mentioned. The following sentence was also added
“Studies have documented various risk factors associated with CIN. These differences show a correlation between CIN and endocrinologic conditions like renal impairment and diabetes. “
Comment 3- Risk factors are quietly repetitive. It would be stronger if grouped conceptually (e.g., baseline kidney function, comorbidities) rather than as a long linear list.
Response- Thank you for pointing this out. We have made the required changes. This has been corrected and is evident in the last part of the second paragraph of the introduction
Comment 4- Some risk factors (e.g., female gender) remain controversial, it might be more appropriate to phrase these as reported in some studies.
Response- Thank you for pointing this out. We have made the required changes. This has been mentioned in the last part of the second paragraph of the introduction
Comment 5- Calling the design retrospective cross-sectional may be confusing, as CIN is inherently a post-exposure outcome. Authors need to clarify it so that the temporal sequence is evident.
Response- Thank you for pointing this out. We have made the required changes. The study design was changed from cross-sectional to prospective longitudinal design to reflect the temporal follow-up after exposure to CECT. The Rationale for the choice of research design was also included.
Comment 6- The choice of 24–72 hours for Scr reassessment is standard but authors need to consider clarifying how many patients had creatinine checked at each time point and whether there was variation in timing that might affect detection of CIN.
Response- Thank you for pointing this out. We have made the required changes. A clarification on the timing has been made in the section 3.
“A majority of participants (85%) had their Scr levels assessed in 48-72 hours after CECT, while the remaining (15%) participants had their levels assessed between 24 and 72 hours after CECT”
Comment 7- The justification for a cross‑sectional design is not accurate for CIN, which requires temporal follow-up after contrast, it should be revised, and the rationale and time window better aligned with the outcome definition.
Response- Thank you for pointing this out. We have made the required changes. The study design was changed from cross-sectional to prospective longitudinal design
Comment 8- When stating that CIN risk may be lower than historically feared, it would be helpful to clarify this as applying to similar low‑risk oncology populations, with preserved renal function, rather than to all cancer patients.
Response- Thank you for pointing this out. We have made the required changes. This has been clarified
“As such, the current study suggests that the risk of CIN among low-risk oncology populations with preserved renal function receiving treatment in Jeddah may be lower than historically feared…”
Reviewer 2 Report
Comments and Suggestions for Authors
The authors present a retrospective cross-sectional study conducted at King Abdulaziz Medical City in Jeddah, evaluating the incidence of Contrast-Induced Nephropathy (CIN) in 80 cancer patients undergoing Contrast-Enhanced Computed Tomography (CECT). There are several issues regarding the draft, as listed below:
- The title implies a general cancer population. However, given the exclusion of patients with baseline Scr > 1.5 mg/dL, the title should be modified to reflect that this study focuses on patients with preserved renal function
- In abstract, the results section mentions "47.71% attrition". It would be helpful to briefly state why nearly half the sample was lost
- The introduction clearly defines CIN using both the relative and absolute increase criteria. However, the study later relies primarily on the percentage increase. It would be best to explicitly justify why the percentage definition was chosen as the primary endpoint over the absolute increase
- In Section 2.2, the manuscript states the endpoint was measured "within 24 to 72 days after CECT". Please clarify. Should it be hours?
- The exclusion of patients with baseline Scr > 1.5 mg/dL creates a selection bias toward "healthy" kidneys. Please discuss it in the limitation section.
- Figure 1: the histogram is helpful, but the x-axis label and bins should be clearer
- In the discussion, the authors attribute the low incidence of CIN to "modern contrast agents with lower osmolarity". Which contrast agent was used?
- The discussion briefly mentions "hydration status" as a limitation. Please elaborate more.
Author Response
Comment 1- The title implies a general cancer population. However, given the exclusion of patients with baseline Scr > 1.5 mg/dL, the title should be modified to reflect that this study focuses on patients with preserved renal function
Response- Thank you for pointing this out. We have made the required changes. The title has been modified to be, “The Incidence of Contrast-induced Nephropathy among Low-Risk Cancer Patients With Preserved Renal Function on Active Treatment Undergoing Contrast-Enhanced Computed Tomography: A single site Experience”
Comment 3- In abstract, the results section mentions "47.71% attrition". It would be helpful to briefly state why nearly half the sample was lost
Response- Thank you for pointing this out. We have made the required changes. An explanation to the high attrition rate is mentioned in the abstract and in the results section.
“The high attrition rate was associated with participants with incomplete records, those who were lost to follow-up, and those whose follow-up Scr was collected after 72 hours from CECT administration”
Comment 3- The introduction clearly defines CIN using both the relative and absolute increase criteria. However, the study later relies primarily on the percentage increase. It would be best to explicitly justify why the percentage definition was chosen as the primary endpoint over the absolute increase
Response- Thank you for pointing this out. We have made the required changes. We have edited the definition of CIN, both in the abstract and introduction sections.
Comment 4- In Section 2.2, the manuscript states the endpoint was measured "within 24 to 72 days after CECT". Please clarify. Should it be hours?
Response- Thank you for pointing this out. We have made the required changes. This has been edited to “hours” instead of days. The sentence now reads”… within 24 to 72 hours after CECT”
“
Comment 5- The exclusion of patients with baseline Scr > 1.5 mg/dL creates a selection bias toward "healthy" kidneys. Please discuss it in the limitation section.
Response- Thank you for pointing this out. We have made the required changes. This limitation has been discussed. The following sentence was added
“. This study is also biased against oncology patients with baseline Scr ≤ 1.5 mg/dL based on the sample selection criteria. As such, these findings may not be applicable among this high-risk group of oncology patients.”
Comment 6- Figure 1: the histogram is helpful, but the x-axis label and bins should be clearer
Response- Thank you for pointing this out. We have made the required changes. We have elaborated on the label. The new label now reads
“A histogram showing post-contrast vs. baseline percentage change in Scr”
Comment 7- In the discussion, the authors attribute the low incidence of CIN to "modern contrast agents with lower osmolarity". Which contrast agent was used?
Response- Thank you for pointing this out. We have made the required changes. We have mentioned iohexol (Omnipaque) in section 2.2 on the third sentence. I however, have not edited the sentence in the discussion since the sentence added in section 2.2 will already have introduced the reader to the contrast agent.
Comment 8- The discussion briefly mentions "hydration status" as a limitation. Please elaborate more.
Response- Thank you for pointing this out. We have made the required changes. The sentence is now elaborate further. It Now Reads
“The study may also suffer residual confounding since it only adjusted for key covariates and ignored other unmeasured factors like chemotherapy timing, volume of contrast, and hydration status that may influence CIN risk by increasing the chances of post-contrast acute kidney injury, raising osmotic load, and reducing renal flow and eGFR , respectively.”
Comment 9- Language, figures and tables can be improved
Response- Thank you for pointing this out. We have made the required changes. Proofreading has been done to improve on the elements mentioned.
Round 2
Reviewer 1 Report
Comments and Suggestions for Authors
The authors have thoroughly addressed all prior comments. I recommend the manuscript for acceptance in the current form without further revision.